# Classification of very low birth weight infants as small for gestational age: International vs. national standards

Marta Tejón-Fernández[1]*, Ana Isabel Armenteros-López[2],
Nazareth Fernández-Rosales[2], Javier Díez-Delgado[2], Diego Salagre[3],
Rafael Galera-Martínez[4], Manuel Martin-González[4], Antonio Bonillo-Perales[5]

1 Department of Biomedical Investigation, Foundation for Biosanitary Research in Eastern Andalusia—FIBAO, Almeria, Spain, 2 Department of Neonatology, Infanta Leonor Women's and Children's Hospital, Almeria, Spain, 3 Department of Pharmacology, BioHealth Institute Granada—IBs Granada, Neuroscience Institute—CIBM, School of Medicine, University of Granada, Granada, Spain, 4 Department of Paediatric Gastroenterology and Nutrition, Infanta Leonor Women's and Children's Hospital, Almeria, Spain, 5 Department of Pneumology, Infanta Leonor Women's and Children's Hospital, Almeria, Spain

* martatejondn@gmail.com

## Abstract

### Background

It is not precisely known whether the use of national (Carrascosa 2008) and international growth standards (INTERGROWTH-21) shows good concordance in classifying very low birth weight infants as small-for-gestational-age or whether with the same degree of morbidity and mortality. The aims of this study were a) to evaluate the concordance between small-for-gestational-age neonates weighing <1500 g classified using the national and international standards, and b) to compare the morbidity and mortality of small-for-gestational-age neonates classified by both standards.

### Methods

A retrospective cohort study was conducted with very low birth weight infants. The concordance between the INTERGROWTH-21 and Carrascosa 2008 standards was analyzed, along with differences in gestational age, weight, medical requirements, morbidity, and mortality, among small-for-gestational-age neonates classified by both standards. Small-for-gestational-age was defined as a birth weight z score ≤ −1.28.

### Results

A total of 250 neonates weighing <1500 g, who were born between 26 and 36 weeks of gestation, were included. There was a high level of concordance in the classification of small-for-gestational-age between the two standards (Cohen's kappa = 0.80, p < 0.001). A lower incidence was observed when the INTERGROWTH-21 standard was used (31.6%) compared to the Carrascosa 2008 standard (40.8%), p = 0.03.

**Data availability statement:** All data files are available from the Figshare database (http://dx.doi.org/10.6084/m9.figshare.27985910).

**Funding:** The author(s) received no specific funding for this work.

**Competing interests:** The authors have declared that no competing interests exist.

No significant differences were found in mortality or morbidity among neonates classified as small-for-gestational-age by both standards.

## Conclusion

The Carrascosa 2008 and INTERGROWTH-21 standards classify small-for-gestational-age infants with comparable morbidity and mortality. We recommend the use of the INTERGROWTH-21 standard for its inclusion of multiple pregnancies, diverse ethnicities, and international comparability.

## Introduction

At present, there is no international consensus about the optimal approach for monitoring the growth of preterm neonates, particularly those with very low birth weight (VLBW) [1]. Neonatologists often face uncertainty about which growth standard to use to classify small-for gestational-age (SGA) preterm neonates, and there is ongoing debate about whether to use national or international standards that include diverse populations to better identify preterm infants who are at risk for complications.

Classifying the growth status of preterm neonates remains a major challenge. Reliable growth standards are essential for identifying SGA preterm neonates, who are known to have significantly higher morbidity and mortality rates. Both national and international growth standards, which were developed based on data from diverse populations, are available for classifying SGA preterm neonates. However, differences in the incidence and prevalence of SGA, as well as in morbidity and mortality outcomes, have been observed depending on the growth standard that is applied. These discrepancies complicate the comparison of results across studies [2–4]. This study contributes to the ongoing debate about the optimal growth charts for evaluating very preterm and extremely preterm neonates, aiming to improve the consistency and comparability of research outcomes in this population [4].

International growth standards, such as INTERGROWTH-21 (INT-21), have been developed specifically for preterm infants. These standards, which were published in 2016, were established based on data from a large multicenter study conducted between 2009 and 2014 in locations including Pelotas, Brazil; Shunyi County, Beijing, China; Central Nagpur, India; Turin, Italy; Parklands Suburb, Nairobi, Kenya; Muscat, Oman; Oxford, United Kingdom; and Seattle, United States. The primary aim of the study was to define the optimal growth patterns for fetuses and neonates. The study population comprised infants who were born between 24 and 36 weeks of gestation whose mothers were healthy, nonsmokers, and free of pregnancy complications such as gestational diabetes. A total of 1,136 preterm neonates were included in the study, representing 5.5% of the total sample. However, the cohort did not include preterm neonates from Spain [5].

In Spain, neonatal growth standards based on gestational age were developed by Carrascosa et al. (Carrascosa 2008) based on data from a prospective multicenter study that was conducted across various Spanish regions. The inclusion criteria focused on neonates who were born between 26 and 42 weeks of gestation from 1999 to 2002 and whose parents were Caucasian and born in Spain. The study included low-risk mothers who were nonsmokers, did not use drugs, experienced uncomplicated pregnancies, and carried healthy fetuses. A total of 1,129 preterm infants were analyzed. These growth standards are widely used by neonatal intensive care unit (NICU) physicians in Spanish hospitals [6].

The aims of this study were a) to evaluate the concordance between the classification of very-low-birth-weight neonates as SGA with the Spanish preterm growth standard (Carrascosa 2008) and with the international growth standard (INT-21), and b) to compare the incidence, morbidity, and mortality rates of SGA neonates in a cohort of Spanish VLBW infants classified using either the international INT-21 growth standard, which was developed based on data from a study that did not include Spanish neonates, or the Spanish preterm growth standard (Carrascosa 2008).

## Methods

Ethics approval for the project was obtained from the Research Ethics Committee of Torrecardenas Hospital (CEI-Almeria code: 9/2017, on March 29th, 2017). Since the study was a retrospective study that used anonymized data, individual patient consent was not required. Access to the data occurred from the 2nd of February 2018 to the 1st of April 2018.

This retrospective cohort study included very low birth weight infants who were admitted to the NICU of Torrecardenas Hospital in Almeria, Spain, which serves a population of 750,000, between 2013 and 2017. The inclusion criteria were VLBW neonates with a gestational age between 26 and 36 weeks (since the Carrascosa 2008 standards do not apply to newborns under 26 weeks of gestation). Infants with genetic disorders; congenital malformations; or endocrine, metabolic, or nutritional diseases not associated with prematurity were excluded.

Data were collected from the computerized medical records, Diraya, and files in the medical archive. Gestational age (GA) was determined based on the date of the mother's last menstrual period. SGA was defined as a birth weight z score $\leq -1.28$, corresponding to the 10th percentile [7]. Weight, length, and head circumference at birth were recorded and categorized using z scores according to the Carrascosa 2008 [6] and INT-21 growth standards [5]. Weight was measured using the precision scale integrated into the incubator (Caleo model, manufactured by Dräger Medical, Germany; catalog code 2M50555-23), which has a measurement accuracy of 1 g and an estimated maximum error of 2–5 g for weights below 2500 g and 5–10 g for weights between 2.5 and 10 kg. Length was measured using the Sayol baby measuring rod, and head circumference was measured using a measuring tape.

The following variables were analyzed: gestational age, birth weight, length, head circumference, Apgar score, bronchopulmonary dysplasia (BPD) [8], necrotizing enterocolitis (NEC) [9], intraventricular hemorrhage (IVH) [10], retinopathy of prematurity (ROP) [11], sepsis, total NICU stay (in days), days of mechanical ventilation, total hospital stay (in days), and mortality.

Statistical analysis was performed using SPSS software, version 23. Cohen's kappa test was used to assess concordance between the Carrascosa 2008 and INT-21 growth standards. Relative risk, the chi-square test, and Student's t test were used to compare the variables. The p value considered statistically significant was 0.05.

## Results

The initial cohort consisted of 289 VLBW infants. After excluding those who did not meet the inclusion and exclusion criteria, as well as preterm infants born between 24 and 26 weeks of gestation (not included in the Carrascosa 2008 standard), a total of 250 VLBW neonates were analyzed. Among this cohort, 52.4% of the infants were female. The mean gestational age of the neonates was $29.69 \pm 2.2$ weeks, with a mean birth weight of $1162.3 \pm 235.8$ g, a mean length of $37.4 \pm 2.8$ cm, and a mean head circumference of $26.8 \pm 2.1$ cm.

The average z scores were compared between the Carrascosa 2008 and INT-21 standards. When the INT-21 standard was compared with the Carrascosa 2008 standard, the z score was significantly greater for weight (−0.77 vs. −0.99, p<0.001), length (−0.87 vs. −1.21, p<0.001) and head circumference (−0.38 vs. −1.21, p<0.001).

A lower percentage of VLBW neonates were classified as SGA when the INT-21 standard was used (n=79, 31.6%) compared to the Carrascosa 2008 standard (n=102, 40.8%; p=0.003). All the neonates who were classified as SGA by INT-21 were also classified as SGA by Carrascosa 2008. However, 23 neonates who were classified as non-SGA by INT-21 were categorized as SGA by Carrascosa 2008. Strong concordance was observed between the two standards (Cohen's kappa=0.80, 95% CI: 0.72–0.87, p<0.001).

The gestational age of the 23 neonates classified as SGA by Carrascosa 2008 but not by INT-21 was similar to the GA of the neonates not classified as SGA according to INT-21 (28.9±1.7 weeks vs. 29.5±2.0 weeks, p=0.10). However, their birth weight was significantly greater (1217±201 g vs. 1094±238 g, p=0.004). No differences were observed in mortality or morbidity outcomes between these groups (p=NS).

No significant differences were found between neonates who were classified as SGA with the INT-21 and Carrascosa 2008 standards in terms of gestational age, birth anthropometry, Apgar scores, NICU stay duration, mechanical ventilation days, parenteral nutrition days, or total hospital stay (p=NS) (Table 1).

No differences were observed in mortality or morbidity among VLBW neonates classified as SGA using the INT-21 standards compared with those classified as SGA using the Carrascosa 2008 standards (Table 2).

**Table 1. Differences in SGA very-low-birth-weight neonates classified with INT-21 and Carrascosa 2008 standards.**

|  | INT-21 (mean±SD) | Carrascosa 2008 (mean±SD) | p value |
|---|---|---|---|
| **Birth Weight (g)** | 1079±265 | 1082±258 | NS |
| **Birth Length (cm)** | 37.1±3.1 | 37.1±3.0 | NS |
| **Gestational Age (weeks)** | 31.2±2.2 | 30.8±2.3 | NS |
| **Apgar at 1 minute** | 7.10±2.7 | 7.09±2.6 | NS |
| **Apgar at 5 minutes** | 8.72±1.96 | 8.75±1.80 | NS |
| **Apgar at 10 minutes** | 9.73±0.86 | 9.75±0.80 | NS |
| **Days of oxygen therapy** | 7.8±19.6 | 9.94±20.8 | NS |
| **NICU stay (days)** | 25.5±27.2 | 25.4±25.0 | NS |

NS, Not significant; NICU, neonatal intensive care unit.

**Table 2. Mortality and morbidity of SGA neonates classified using INT-21 and Carrascosa 2008 standards.**

| Risk factor | SGA INT-21 | SGA Carrascosa 2008 | RR | p value |
|---|---|---|---|---|
| **Mortality 28 days** | 11.4% | 9.8% | 1.16 | NS |
| **Total mortality** | 13.9% | 11.8% | 1.18 | NS |
| **BPD** | 6.5% | 10% | 0.64 | NS |
| **ROP** | 12.3% | 14.3% | 0.86 | NS |
| **IVH** | 23.5% | 27.3% | 0.85 | NS |
| **NEC** | 10.1% | 9.2% | 1.10 | NS |
| **Early-onset sepsis** | 8.5% | 6.5% | 1.31 | NS |
| **Late-onset sepsis** | 19.4% | 22.5% | 0.85 | NS |

SGA, small-for-gestational-age; RR, relative risk; BPD, bronchopulmonary dysplasia; ROP, retinopathy of prematurity; IVH, intraventricular hemorrhage; NEC, necrotizing enterocolitis; NS, not significant.

## Discussion

In countries that use the INT-21 standard as the growth standard, the rates of small-of-gestational-age among very low birth weight neonates range from 25% to 35% [1,4,12,13]. The differences in the incidence of SGA among different studies may occur due to the inclusion of different populations of pregnant women. In our study, we did not exclude pregnant women with gestational diabetes or eclampsia, whereas these conditions were excluded in the INT-21 study [5].

Although Spanish preterm neonates were not included in the study according to which the INT-21 standard was developed, our study revealed a prevalence of 31.6%, which was identical to that reported in a recent Spanish national multi-center study that analyzed 23,582 VLBW neonates and reported a prevalence of 31.1% [14]. Therefore, we consider that INT-21 can be reliably used in Spain to classify these preterm neonates with the same accuracy as in the countries that participated in the development of INT-21 [3,12,13,15].

We observed a high degree of concordance between the INT-21 and Carrascosa 2008 standards in classifying VLBW neonates as small for gestational age (SGA) (Kappa: 0.80) [1,3,4,16]. Despite this high concordance (both standards classified 91.8% of preterm neonates identically), we found significant differences in the percentage of infants who were classified as SGA between the Carrascosa 2008 and INT-21 standards (40.8% vs. 31.6%, p < 0.001). These differences may be due to the fact that the Carrascosa 2008 standard was developed based on a study that included only Caucasian newborns with parents born in Spain and excluded all preterm infants from multiple gestations.

The discrepancies between the INT-21 and Carrascosa 2008 standards corresponded to 23 preterm newborns who were classified as SGA according to the Carrascosa 2008 standard but not according to the INT-21 standard. These preterm newborns presented a lower birth weight but the same gestational age as those classified as non-SGA according to INT-21. These discrepancies could occur because the Carrascosa 2008 standard was developed based on data from a study that excluded multiple gestations. Furthermore, the morbidity and mortality of SGA newborns are similar to those of newborns classified as non-SGA according to the INT-21 standards, which contributes to the lack of differences in morbidity and mortality among SGA newborns, regardless of whether they were classified using the INT-21 or Carrascosa 2008 standard (Tables 1 and 2) [2,16–18].

This study is subject to certain limitations. The retrospective design of this study introduces potential information biases, as some data were unavailable from the patients' medical records. Additionally, prenatal and perinatal complications, maternal illnesses, and intrauterine growth factors were not analyzed, despite their potential effects on neonatal growth outcomes. These limitations should be carefully considered in the context of the findings and addressed in future research endeavors.

## Conclusions

The percentage of very low birth weight neonates classified as SGA according to the INT-21 standard is similar to the percentages classified as SGA in other Spanish studies and to the percentages reported in the countries that participated in the INT-21 study. Therefore, the INT-21 standard is suitable for comparing the results of different SGA studies at both the national and international levels.

In Spanish preterm neonates, we observed strong concordance between the INT-21 and Carrascosa 2008 standards for SGA classification. However, we recommend the use of the INT-21 standard, as the Carrascosa-2008 standard was developed based on a study that only analyzed Caucasian preterm neonates born to Spanish parents and excluded multiple pregnancies. This limitation may reduce its accuracy in studying non-Caucasian preterm neonates and those from multiple pregnancies.

Large-scale, prospective, multicenter studies are necessary to enhance the classification and analysis of morbidity among VLBW neonates who are small-for-gestational-age with greater precision.

## Author contributions

**Conceptualization:** Marta Tejón-Fernández, Rafael Galera-Martínez.

**Data curation:** Marta Tejón-Fernández, Ana Isabel Armenteros-López.

**Formal analysis:** Nazareth Fernández-Rosales, Antonio Bonillo-Perales.

**Methodology:** Rafael Galera-Martínez, Antonio Bonillo-Perales.

**Supervision:** Javier Díez-Delgado, Manuel Martin-González, Antonio Bonillo-Perales.

**Writing – original draft:** Marta Tejón-Fernández, Ana Isabel Armenteros-López.

**Writing – review & editing:** Diego Salagre, Manuel Martin-González.

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
