## [Decision Letter · Decision Letter 0]

22 Nov 2024

PONE-D-24-41520Small-for-gestational-age in very low birth weight: International vs. national standardsPLOS ONE

Dear Dr. Tejón Fernández,

Thank you for submitting your manuscript to PLOS ONE. After careful consideration, we feel that it has merit but does not fully meet PLOS ONE’s publication criteria as it currently stands. Therefore, we invite you to submit a revised version of the manuscript that addresses the points raised during the review process.

We look forward to receiving your revised manuscript.

Kind regards,

Tamara Sljivancanin Jakovljevic

Academic Editor

PLOS ONE

**Journal Requirements:**

2. In the online submission form, you indicated that All data underlying the results presented in the study are available from Marta Tejón-Fernández (martatejondn@gmail.com), who is the first and corresponding author.

Reviewers' comments:

Reviewer's Responses to Questions

**Comments to the Author**

1. Is the manuscript technically sound, and do the data support the conclusions?

Reviewer #1: Partly

Reviewer #2: Yes

2. Has the statistical analysis been performed appropriately and rigorously? 

Reviewer #1: Yes

Reviewer #2: Yes

3. Have the authors made all data underlying the findings in their manuscript fully available?

Reviewer #1: Yes

Reviewer #2: Yes

4. Is the manuscript presented in an intelligible fashion and written in standard English?

Reviewer #1: No

Reviewer #2: No

5. Review Comments to the Author

**Reviewer #1:**  Fernández and co-authors have compared demographic characteristics and outcomes (mortality and several morbidities) in small-for-gestational age (SGA) babies, selected from a preterm low birth weight population of 250 babies born in a single hospital in Spain, according to whether they were classified as SGA according to national or international birth weight distributions. The analysis involved retrieval of electronic patient records and hardcopy files to obtain information on outcomes, then comparison of outcomes between SGA groups defined by the two classifications. Only five years of data, generating a cohort of 250 babies, was used, and perhaps a larger cohort would have allowed for more rigorous testing of whether rarer outcomes such as sepsis differ depending on SGA classification cut-offs. Clarification of the question and discussion of the implications would improve the manuscript. The English usage is hard to understand in places.

Specific comments:

a. Title page – better to provide an institutional Email as contact, rather than gmail

b. Abstract – this needs additional information, e.g. upper and lower gestational age boundary for the cohort (not just lower limit).

c. English usage – needs correction/clarification in places, e.g.

• Spanish not Spaniard in abstract.

• Data are all quantitative not qualitative (statistics in methods – just not all continuous/normally distributed).

• Abstract: Differences in outcomes are not between growth charts, but between babies classified according to the two different systems/sets of cut-offs.

• Line 62: use the “less than or equal symbol” to make this clearer

• Line 65 “being 15% of this percent less than…” – very unclear.

• Line 123 babies are male or female, not men or women.

• Line 128-129 – what do you mean by minor? Different or not?

• Line 147 “it also did not observe differences between” - unclear

d. Introduction – much of this section was hard to follow. The initial sentences were about prematurity, not classification systems or why this might be important. Lines 57-58 seems a mixed argument comparing VLBW with late preterm not normally grown? The question of classification standards and what has been done previously was not reviewed until the discussion – I recommend reworking the introduction to cover this material and to clearly explain the gap in understanding that the present study addresses. Why did you focus on very preterm babies (26-32 weeks) only? Line 68 – what is EEUU? Define abbreviations. Line 87 – what do you mean by “adequate pregnancy?” I think the sentence at lines 72-74 is intended as a statement of the rationale for the present study, but it is very hard to follow. The statement of the aim (lines 90-93) is confusing; could you split this into two parts?

e. Methods – do you have access to other important outcomes for this preterm cohort, e.g. need for supplemental oxygen at discharge? Respiratory distress syndrome? Give manufacturer and city in brackets not text – is there also a catalog number for the incubator that was used?

f. Ethics – this is usually placed at the start of the methods.

g. Results – text should be revised to present data only to meaningful levels of precision. For example, the methods state that the birth weight of babies could be measured to the nearest gram, so weight should be presented as whole grams only. Similarly, if you are measuring head circumference with a tape, the precision is not going to be greater than 0.1 cm at best. The level of precision in the tables is better, although editing is needed to ensure that all means and standard deviations for a given outcome are described with the same number of decimal places.

h. Figure 1 – I suggest removing Figure 1, because its’ contents are very unclear and it is a poor-quality figure. Instead, you could provide the actual data and z-scores for average weight, crown-rump length and head circumferences for the cohort according to each classification system in a Table. Providing mean plus/minus standard deviation for z-scores would be more informative than only mean. I also recommend that you shift the sentences on gestational age from the end of the results into this first section of results – this data could also be added to an initial “demographics” table, which could include the z-scores for the cohort of babies according to each classification system.

i. Table 1 is not needed – just place the numbers and percentages in text and avoid repeating these.

j. Tables 2 and 3 – it would be more logical to reorganise these so that demographic information is in one table and all of the outcomes related to morbidity (including NICU admission, supplemental oxygen, BPD, IVH etc) are in the other.

k. Table 4 and 5 are not relevant to your question and could be removed. Differences between AGA and SGA are well known, not novel and not related to your question about whether outcomes differ according to which classification system is used to define SGA. That question is already addressed in Table 3.

l. Discussion – This section would be improved by rewriting to clearly discuss the findings of the present study and its implications. Descriptions of what others have found when comparing classification systems belongs in the introduction to clearly explain what gap in knowledge you are addressing. These also need to be better integrated to explain the overall argument as this currently reads as a series of paragraphs each describing a single study, which are not clearly linked.

m. Conclusions – these are fairly limited. Can you make recommendations about which classification systems should be used in Spain based on your study? Is it better to use the local growth charts and define more babies as SGA or the international standard and define fewer babies as SGA. Does this impact the clinical care or decisions made about theses babies? Would using local standards limit ability to compare studies internationally? Are your findings relevant to other populations, or only in the Spanish context?

**Reviewer #2: ** The rationale for the study is justified since Spanish population was not part of the intergrowth 21 data set. Statistical methodology is also acceptable. The authors should consider correcting multiple grammatical errors before resubmission.

6. PLOS authors have the option to publish the peer review history of their article (what does this mean? ). If published, this will include your full peer review and any attached files.

**Do you want your identity to be public for this peer review?** For information about this choice, including consent withdrawal, please see our Privacy Policy .

Reviewer #1: No

Reviewer #2: No

---

## [Author Response · Author response to Decision Letter 0]

8 Dec 2024

Journal Requirements:

Please ensure that your manuscript meets PLOS ONE's style requirements, including those for file naming. The PLOS ONE style templates can be found at https://journals.plos.org/plosone/s/file?id=wjVg/PLOSOne_formatting_sample_main_body.pdf and https://journals.plos.org/plosone/s/file?id=ba62/PLOSOne_formatting_sample_title_authors_affiliations.pdf

Authors' response: The revised manuscript has been prepared in accordance with the journal's style requirements.

2. In the online submission form, you indicated that All data underlying the results presented in the study are available from Marta Tejón-Fernández (martatejondn@gmail.com), who is the first and corresponding author.

Authors' response: The data have been uploaded to the Figshare repository (http://dx.doi.org/10.6084/m9.figshare.27985910).

1. Is the manuscript technically sound, and do the data support the conclusions?

Reviewer #1: Partly

Reviewer #2: Yes

Authors' response: Thank you, reviewers. In response to Reviewer #1’s comment, in the manuscript, we have included comments regarding the limitation related to the sample size. Furthermore, we have undertaken a revision of the conclusions to ensure their alignment with the stated objectives, substantiating them with the results obtained.

2. Has the statistical analysis been performed appropriately and rigorously?

Reviewer #1: Yes

Reviewer #2: Yes

3. Have the authors made all data underlying the findings in their manuscript fully available?

Reviewer #1: Yes

Reviewer #2: Yes

4. Is the manuscript presented in an intelligible fashion and written in standard English?

Reviewer #1: No

Reviewer #2: No

Authors' response: The manuscript has undergone a comprehensive review to address typographical and grammatical errors. We have improved the manuscript's writing to ensure its adaptation to standard English.

5. Review Comments to the Author

Reviewer #1: Fernández and co-authors have compared demographic characteristics and outcomes (mortality and several morbidities) in small-for-gestational age (SGA) babies, selected from a preterm low birth weight population of 250 babies born in a single hospital in Spain, according to whether they were classified as SGA according to national or international birth weight distributions. The analysis involved retrieval of electronic patient records and hardcopy files to obtain information on outcomes, then comparison of outcomes between SGA groups defined by the two classifications. Only five years of data, generating a cohort of 250 babies, was used, and perhaps a larger cohort would have allowed for more rigorous testing of whether rarer outcomes such as sepsis differ depending on SGA classification cut-offs. Clarification of the question and discussion of the implications would improve the manuscript. The English usage is hard to understand in places.

Authors' response: Thank you, Reviewer, for your valuable comments. We have included a discussion in the manuscript regarding the limitations of the sample size, along with suggestions for addressing this issue in future research. Additionally, the manuscript has been thoroughly reviewed to correct grammatical errors and ensure its adaptation to standard English.

Specific comments:

a. Title page – better to provide an institutional Email as contact, rather than gmail.

Authors' response: At present, Tejón-Fernández does not have an institutional Email.

b. Abstract – this needs additional information, e.g. upper and lower gestational age boundary for the cohort (not just lower limit).

Authors' response: The upper limit of gestational age has been incorporated into the abstract.

“250 neonates weighing <1500 g, born between 26 and 36 weeks of gestation were included.”

c. English usage – needs correction/clarification in places, e.g.

• Spanish not Spaniard in abstract.

Authors' response: Corrected.

• Data are all quantitative not qualitative (statistics in methods – just not all continuous/normally distributed).

Authors' response: Corrected, the Chi-squared test and Student's t-test were used for variable comparison.

• Abstract: Differences in outcomes are not between growth charts, but between babies classified according to the two different systems/sets of cut-offs.

Authors' response: The suggested corrections have been implemented.

• Line 62: use the “less than or equal symbol” to make this clearer.

Authors' response: Corrected.

• Line 65 “being 15% of this percent less than…” – very unclear.

Authors' response: As suggested by the reviewers, that paragraph is unclear and contributes little to the manuscript; therefore, the comment has been removed.

• Line 123 babies are male or female, not men or women.

Authors' response: The text has been corrected.

“Of this cohort, 52.4% were female”.

• Line 128-129 – what do you mean by minor? Different or not?

Authors' response: This has been clarified and corrected in the text.

“The average z-scores were compared between the Carrascosa 2008 and INT-21 standards. When using the INT-21 standards compared to Carrascosa 2008 the z-score was statistically significantly higher for weight (−0.77 vs. −0.99, p < 0.001), length (−0.87 vs. −1.21, p < 0.001) and head circumference (−0.38 vs. −1.21, p < 0.001).”

• Line 147 “it also did not observe differences between” – unclear.

Authors' response: The text has been revised to enhance clarity.

“No significant differences were found between neonates classified as SGA using INT-21 and Carrascosa 2008 standards regarding gestational age, birth anthropometry, Apgar scores, NICU stay duration, mechanical ventilation days, parenteral nutrition days, or total hospital stay (p = NS)(Table 1).”

d. Introduction – much of this section was hard to follow. The initial sentences were about prematurity, not classification systems or why this might be important.

Authors' response: Grammatical corrections have been made, along with the implementation of the proposed changes.

Lines 57-58 seems a mixed argument comparing VLBW with late preterm not normally grown? The question of classification standards and what has been done previously was not reviewed until the discussion – I recommend reworking the introduction to cover this material and to clearly explain the gap in understanding that the present study addresses.

Authors' response: The corrections in the introduction have been made based on the reviewer' suggestions.

Why did you focus on very preterm babies (26-32 weeks) only?

Authors' response: There was a transcription error: the preterm neonates in our study are within the range of 26 to 36 weeks of gestation. The choice of 26 weeks was based on the fact that the Carrascosa 2008 standards were established using a cohort starting at 26 weeks of gestation. These details have been corrected and included in the manuscript.

Line 68 – what is EEUU? Define abbreviations.

Authors' response: Corrected.

“Regarding SGA neonates, a cohort of infants weighing less than 1500 grams (g) in the United States in 2021…”.

Line 87 – what do you mean by “adequate pregnancy?”

Authors' response: It has been clarified in the text.

“The study included low-risk mothers who were nonsmokers, did not use drugs, experienced uncomplicated pregnancies…”.

I think the sentence at lines 72-74 is intended as a statement of the rationale for the present study, but it is very hard to follow.

Authors' response: The rationale for the study has been clarified, and one of the bibliographic references has been modified to enhance understanding.

15. Rajendra Prasad A, Venkateshwarlu V, · Srinivas M, Sai Kiran D and· Tejo Pratap O. Comparison of Fenton, INTERGROWTH 21st, and Population Based Growth Charts in Predicting Outcomes of Very Preterm Small for Gestational Age Neonates. Indian Journal of Pediatrics. 2022; 89(10): 1034–36. https://doi.org/10.1007/s12098-022-04175-3 PMID: 35604586

The statement of the aim (lines 90-93) is confusing; could you split this into two parts?

Authors' response: The aims of the study have been clarified in two sections.

“The aims of this study were a) to evaluate the concordance between SGA very low birth weight neonates classified using the Spanish preterm growth standard (Carrascosa 2008) and those classified using the international growth standard (INT-21); b) to compare the incidence, morbidity, and mortality rates of SGA neonates in a cohort of Spanish VLBW infants, classified using either the international INT-21 growth standard, which does not include Spanish neonates, or the Spanish preterm growth standard (Carrascosa 2008).”.

e. Methods – do you have access to other important outcomes for this preterm cohort, e.g. need for supplemental oxygen at discharge? Respiratory distress syndrome?

Authors' response: Only the days of supplemental oxygen were recorded.

Give manufacturer and city in brackets not text – is there also a catalog number for the incubator that was used?

Authors' response: The incubator used was the Caleo model, manufactured by Dräger Medical, catalog code 2M50555-23.

f. Ethics – this is usually placed at the start of the methods.

Authors' response: The ethical considerations have been included at the beginning of the Methods section, as suggested by the reviewers.

“Ethics approval for the project was obtained from the Research Ethics Committee of Torrecardenas Hospital (CEI-Almeria code: 9/2017, on March 29th, 2017). Since the study was a retrospective study using anonymized data, individual patient consent was not required. Access to the data was on the 2nd of February 2018.”

g. Results – text should be revised to present data only to meaningful levels of precision. For example, the methods state that the birth weight of babies could be measured to the nearest gram, so weight should be presented as whole grams only. Similarly, if you are measuring head circumference with a tape, the precision is not going to be greater than 0.1 cm at best. The level of precision in the tables is better, although editing is needed to ensure that all means and standard deviations for a given outcome are described with the same number of decimal places.

Authors' response: The results have been revised based on the improvement suggestions provided by the reviewers.

h. Figure 1 – I suggest removing Figure 1, because its’ contents are very unclear and it is a poor-quality figure. Instead, you could provide the actual data and z-scores for average weight, crown-rump length and head circumferences for the cohort according to each classification system in a Table. Providing mean plus/minus standard deviation for z-scores would be more informative than only mean. I also recommend that you shift the sentences on gestational age from the end of the results into this first section of results – this data could also be added to an initial “demographics” table, which could include the z-scores for the cohort of babies according to each classification system.

Authors' response: Figure 1 has been removed, and the corresponding results have been specified more precisely in the text.

i. Table 1 is not needed – just place the numbers and percentages in text and avoid repeating these.

Authors' response: Table 1 has been removed, and the results are now presented in the text.

j. Tables 2 and 3 – it would be more logical to reorganise these so that demographic information is in one table and all of the outcomes related to morbidity (including NICU admission, supplemental oxygen, BPD, IVH etc) are in the other.

Authors' response: The modifications suggested by the reviewers have been implemented.

k. Table 4 and 5 are not relevant to your question and could be removed. Differences between AGA and SGA are well known, not novel and not related to your question about whether outcomes differ according to which classification system is used to define SGA.

Authors' response: That question is already addressed in actual Table 2. Tables 4 and 5 have been removed as suggested by the reviewers.

l. Discussion – This section would be improved by rewriting to clearly discuss the findings of the present study and its implications. Descriptions of what others have found when comparing classification systems belongs in the introduction to clearly explain what gap in knowledge you are addressing. These also need to be better integrated to explain the overall argument as this currently reads as a series of paragraphs each describing a single study, which are not clearly linked.

Authors' response: Based on the improvement suggestions and questions raised by the reviewers, we have revised the Discussion section and conducted the following statistical analysis, which has been added to the Results section. This analysis allows us to address most of the questions posed by the reviewers.

“The gestational age of the 23 neonates classified as SGA by Carrascosa 2008 but not by INT-21 was similar to the GA of the non-SGA group according to INT-21 (28.9 ± 1.7 weeks vs. 29.5 ± 2.0 weeks, p = 0.10). However, their birth weight was statistically significantly higher (1217 ± 201 g vs. 1094 ± 238 g, p = 0.004). No differences were observed in mortality or morbidity outcomes between these groups (p = NS).”

The Discussion section has been revised to incorporate the considerations suggested by the reviewers.

m. Conclusions – these are fairly limited.

Can you make recommendations about which classification systems should be used in Spain based on your study?

Authors' response: The comment has been incorporated into the Discussion and Conclusions sections.

“Both standards classify preterm infants with comparable neonatal morbidity and mortality, making it reasonable to recommend the use of either Carrascosa 2008 or INT-21 for classifying SGA preterm neonates in Spain.”

Is it better to use the local growth charts and define more babies as SGA or the international standard and define fewer babies as SGA?

Authors' response: The comment has been included in the Discussion section.

Does this impact the clinical care or d

---

## [Decision Letter · Decision Letter 1]

26 Jan 2025

PONE-D-24-41520R1Small-for-gestational-age in very low birth weight: International vs. national standardsPLOS ONE

Dear Dr. Tejón Fernández,

Thank you for submitting your manuscript to PLOS ONE. After careful consideration, we feel that it has merit but does not fully meet PLOS ONE’s publication criteria as it currently stands. Therefore, we invite you to submit a revised version of the manuscript that addresses the points raised during the review process.

We look forward to receiving your revised manuscript.

Kind regards,

Tamara Sljivancanin Jakovljevic

Academic Editor

PLOS ONE

Reviewers' comments:

Reviewer's Responses to Questions

**Comments to the Author**

1. If the authors have adequately addressed your comments raised in a previous round of review and you feel that this manuscript is now acceptable for publication, you may indicate that here to bypass the “Comments to the Author” section, enter your conflict of interest statement in the “Confidential to Editor” section, and submit your "Accept" recommendation.

Reviewer #1: (No Response)

Reviewer #2: (No Response)

2. Is the manuscript technically sound, and do the data support the conclusions?

Reviewer #1: Yes

Reviewer #2: Yes

3. Has the statistical analysis been performed appropriately and rigorously? 

Reviewer #1: Yes

Reviewer #2: Yes

4. Have the authors made all data underlying the findings in their manuscript fully available?

Reviewer #1: Yes

Reviewer #2: Yes

5. Is the manuscript presented in an intelligible fashion and written in standard English?

Reviewer #1: No

Reviewer #2: No

6. Review Comments to the Author

Reviewer #1: Thank you for addressing many of the comments made on the original manuscript, including clarification of the aims statement and substantial revision of the results Tables. There are some areas where I think the manuscript could still be improved, and/or further changes are needed to appropriately address the comments. In some cases the information was provided in responses but not incorporated into the actual manuscript. Please see points below. Could you please indicate the page and line of the tracked changes version where further changes are made, to make re-review faster?

• The English usage/wording is still unclear in places, e.g. first sentence of abstract (not a complete sentence and meaning is unclear): “The comparability of the Spanish neonatal growth standards (Carrascosa 2008) and the international INTERGROWTH-21 standards in classifying very low birth weight neonates with the same morbidity and mortality rates as small-for-gestational-age is unknown, and what the degree of concordance between these standards is.”

• I also suggest reworking the introduction further to clearly introduce the question of the present study. It is not clear why the introduction starts with a discussion of VLBW and then a section on incidence on preterm birth, when the question this study addresses is about SGA classification and impact. The introduction relevant to this study is not stated until much later (sentence starting “classifying the growth status”). A point for the authors to note is that not all SGA are preterm.

• The rationale for using a cohort between 26 and 36 weeks should be added to the manuscript.

• Incubator model and manufacturer location that were provided in the response need to be incorporated in the manuscript.

• Thank you for revising the Tables. The point about precision of results, and ensuring means and errors are reported to the same number of decimal places needs further attention, as this has been missed in places, e.g. line 187 in tracked changes manuscript.

• The point added at line 271 (Neonatologists often face uncertainty about which growth standard to use for classifying SGA preterm neonates, and there is ongoing debate about whether to use national standards or international ones that encompass diverse populations to better identify preterm at risk for complications.”) provides good rationale for the present study and would work well in the introduction), but does not provide a conclusion or recommendation based on the findings of this study and of previous literature, which would be expected in a discussion section. It also does not address the reviewer query: “Are your findings relevant to other populations, or only in the Spanish context?”! Can the findings of THIS study be generalised to other populations, or are they only important for classifying Spanish babies?

Reviewer #2: Authors have diligently addressed the points raised in the rebuttal letter. However, a few improvements are necessary to bring the submission to an acceptable standard:

• Editing the grammatical errors in lines 31-32 and 66-70 will enhance readability.

• The claim of accessing data in just one day (Feb 2nd, 2018) seems exceptionally efficient, almost reaching implausible limits. It may be beneficial to either provide a plausible range or leave it unchanged if accurate.

• The inclusion criteria for INT-21 were based on maternal characteristics (lines 85-87) and differ from those chosen by the authors (lines 110-112). While the patient populations may end up similar in practical terms, an explanation for this deviation is essential. This deviation could potentially explain the differences in outcomes.

• The Z scores for weight, length, and head circumference show statistically significant differences, with a substantial observed difference of an additional 23 neonates classified as SGA using Carrascosa 2008. Although I concur with the calculated concordance using Cohen’s kappa statistic, providing additional statistical rationale could enhance reader acceptance.

7. PLOS authors have the option to publish the peer review history of their article (what does this mean? ). If published, this will include your full peer review and any attached files.

**Do you want your identity to be public for this peer review?** For information about this choice, including consent withdrawal, please see our Privacy Policy .

Reviewer #1: No

Reviewer #2: No

---

## [Author Response · Author response to Decision Letter 1]

3 Mar 2025

We have included M. Martín-González as one of the authors of the manuscript. Given their extensive expertise, they have provided invaluable assistance in restructuring the article based on the requested modifications.

The title of the manuscript has been revised following the recommendation of AJE, the editing service commissioned to refine the English language usage, as they indicated that a correction was necessary.

Author's Responses to Reviewers’ questions:

1. If the authors have adequately addressed your comments raised in a previous round of review and you feel that this manuscript is now acceptable for publication, you may indicate that here to bypass the “Comments to the Author” section, enter your conflict of interest statement in the “Confidential to Editor” section, and submit your "Accept" recommendation.

Reviewer #1: (No Response)

Reviewer #2: (No Response)

2. Is the manuscript technically sound, and do the data support the conclusions?

Reviewer #1: Yes

Reviewer #2: Yes

3. Has the statistical analysis been performed appropriately and rigorously?

Reviewer #1: Yes

Reviewer #2: Yes

4. Have the authors made all data underlying the findings in their manuscript fully available?

Reviewer #1: Yes

Reviewer #2: Yes

5. Is the manuscript presented in an intelligible fashion and written in standard English?

Reviewer #1: No

Reviewer #2: No

The language-related corrections have been carried out by a professional editing company, AJE, which has revised and edited the text to ensure its adherence to the appropriate linguistic standards. The certification is included in the documents submitted to the journal.

6. Review Comments to the Author

Reviewer #1: Thank you for addressing many of the comments made on the original manuscript, including clarification of the aims statement and substantial revision of the results Tables. There are some areas where I think the manuscript could still be improved, and/or further changes are needed to appropriately address the comments. In some cases the information was provided in responses but not incorporated into the actual manuscript. Please see points below. Could you please indicate the page and line of the tracked changes version where further changes are made, to make re-review faster?

Thank you very much for all your contributions. Although some paragraphs have been removed and others have been substantially modified, In response to your request, we have enabled line numbers in the track changes document. This will allow you to view all modifications made to the manuscript, along with their corresponding justifications.

• The English usage/wording is still unclear in places, e.g. first sentence of abstract (not a complete sentence and meaning is unclear): “The comparability of the Spanish neonatal growth standards (Carrascosa 2008) and the international INTERGROWTH-21 standards in classifying very low birth weight neonates with the same morbidity and mortality rates as small-for-gestational-age is unknown, and what the degree of concordance between these standards is.”

The English usage/wording has been reviewed by AJE, this sentence has been completely modified.

Lines 34-39 (page 2/18): “It is not precisely known whether the use of national (Carrascosa 2008) and international growth standards (INTERGROWTH-21) shows good concordance in classifying very low birth weight infants as small-for-gestational-age or whether with the same degree of morbidity and mortality.”

• I also suggest reworking the introduction further to clearly introduce the question of the present study. It is not clear why the introduction starts with a discussion of VLBW and then a section on incidence on preterm birth, when the question this study addresses is about SGA classification and impact. The introduction relevant to this study is not stated until much later (sentence starting “classifying the growth status”). A point for the authors to note is that not all SGA are preterm.

Revisions have been made to the text in accordance with the request, ensuring a clear introduction of the research question (page 3-5/18).

• The rationale for using a cohort between 26 and 36 weeks should be added to the manuscript.

The justification has been added to the Methods section.

Lines 132-134 (page 6/18): “The inclusion criteria were VLBW neonates with a gestational age between 26 and 36 weeks (since the Carrascosa 2008 standards do not apply to newborns under 26 weeks of gestation).”

• Incubator model and manufacturer location that were provided in the response need to be incorporated in the manuscript.

You are right. We apologize for the oversight, as we did not include the incubator model in the manuscript during the previous revision.

Lines 141-142 (page 6/18): “Weight was measured using the precision scale integrated into the incubator (Caleo model, manufactured by Dräger Medical, Germany; catalog code 2M50555-23)…”

• Thank you for revising the Tables. The point about precision of results, and ensuring means and errors are reported to the same number of decimal places needs further attention, as this has been missed in places, e.g. line 187 in tracked changes manuscript.

The correction has been made, keeping one decimal place for height and head circumference, as you suggested.

Lines 158-160 (page 7/18): “The mean gestational age of the neonates was 29.69 ± 2.2 weeks, with a mean birth weight of 1162.3 ± 235.8 g, a mean length of 37.4 ± 2.8 cm, and a mean head circumference of 26.8 ± 2.1 cm.”

• The point added at line 271 (Neonatologists often face uncertainty about which growth standard to use for classifying SGA preterm neonates, and there is ongoing debate about whether to use national standards or international ones that encompass diverse populations to better identify preterm at risk for complications.”) provides good rationale for the present study and would work well in the introduction), but does not provide a conclusion or recommendation based on the findings of this study and of previous literature, which would be expected in a discussion section. It also does not address the reviewer query: “Are your findings relevant to other populations, or only in the Spanish context?”! Can the findings of THIS study be generalised to other populations, or are they only important for classifying Spanish babies?

We have addressed your valuable comments in the discussion section, further elaborating on the justification for the study. Additionally, we have refined the conclusions to reflect the insightful points raised during the review process (page 9-12/18).

Reviewer #2: Authors have diligently addressed the points raised in the rebuttal letter.

We greatly appreciate your comments. As requested by Reviewer #1, we have included line numbers and page in the track changes document to facilitate your re-review process.

However, a few improvements are necessary to bring the submission to an acceptable standard:

• Editing the grammatical errors in lines 31-32 and 66-70 will enhance readability.

The grammatical errors highlighted have been corrected and reviewed by AJE, the editing service we commissioned to enhance the English language usage. We have attached the certification provided by AJE for the journal's reference.

• The claim of accessing data in just one day (Feb 2nd, 2018) seems exceptionally efficient, almost reaching implausible limits. It may be beneficial to either provide a plausible range or leave it unchanged if accurate.

Access to the data was granted for a period of two months, this has been included in the manuscript.

Lines 126-129 (page 5-6/18): “Access to the data occurred from the 2nd of February 2018 to the 1st of April 2018.”

• The inclusion criteria for INT-21 were based on maternal characteristics (lines 85-87) and differ from those chosen by the authors (lines 110-112). While the patient populations may end up similar in practical terms, an explanation for this deviation is essential. This deviation could potentially explain the differences in outcomes.

The comment has been incorporated into the Discussion section.

Lines 198-202 (page 9/18): “In countries that use the INT-21 standard as the growth standard, the rates of small-of-gestational-age among very low birth weight neonates range from 25% to 35% [1,4,12,14]. The differences in the incidence of SGA among different studies may occur due to the inclusion of different populations of pregnant women. In our study, we did not exclude pregnant women with gestational diabetes or eclampsia, whereas these conditions were excluded in the INT-21 study [5].”

• The Z scores for weight, length, and head circumference show statistically significant differences, with a substantial observed difference of an additional 23 neonates classified as SGA using Carrascosa 2008. Although I concur with the calculated concordance using Cohen’s kappa statistic, providing additional statistical rationale could enhance reader acceptance.

The Discussion now includes the finding that the agreement rate for classifying SGA according to the two standards is 91.8%, which underscores the high level of concordance. However, all discrepancies are attributed to 23 newborns, all of whom were classified as SGA according to Carrascosa 2008 but none according to INT-21. This accounts for the differences in the incidence rates of SGA.

7. PLOS authors have the option to publish the peer review history of their article (what does this mean?). If published, this will include your full peer review and any attached files.

Do you want your identity to be public for this peer review? For information about this choice, including consent withdrawal, please see our Privacy Policy.

Reviewer #1: No

Reviewer #2: No

---

## [Decision Letter · Decision Letter 2]

1 Apr 2025

PONE-D-24-41520R2Classification of very low birth weight infants as small for gestational age: International vs. national standards.PLOS ONE

Dear Dr. Tejón Fernández,

Thank you for submitting your manuscript to PLOS ONE. After careful consideration, we feel that it has merit but does not fully meet PLOS ONE’s publication criteria as it currently stands. Therefore, we invite you to submit a revised version of the manuscript that addresses the points raised during the review process.

We look forward to receiving your revised manuscript.

Kind regards,

Tamara Sljivancanin Jakovljevic

Academic Editor

PLOS ONE

**Journal Requirements:**

**Additional Editor Comments:**

Dear Authors,

please make few minor changes regarding text style, requested by the reviewers. After their implementation, your manuscript can be accepted for the publication.

Reviewers' comments:

Reviewer's Responses to Questions

**Comments to the Author**

1. If the authors have adequately addressed your comments raised in a previous round of review and you feel that this manuscript is now acceptable for publication, you may indicate that here to bypass the “Comments to the Author” section, enter your conflict of interest statement in the “Confidential to Editor” section, and submit your "Accept" recommendation.

Reviewer #1: All comments have been addressed

Reviewer #2: (No Response)

2. Is the manuscript technically sound, and do the data support the conclusions?

Reviewer #1: (No Response)

Reviewer #2: Yes

3. Has the statistical analysis been performed appropriately and rigorously? 

Reviewer #1: (No Response)

Reviewer #2: Yes

4. Have the authors made all data underlying the findings in their manuscript fully available?

Reviewer #1: (No Response)

Reviewer #2: Yes

5. Is the manuscript presented in an intelligible fashion and written in standard English?

Reviewer #1: (No Response)

Reviewer #2: Yes

6. Review Comments to the Author

**Reviewer #1:**  (No Response)

**Reviewer #2: ** The authors have taken good efforts to address most of the issues. Reasoning behind results have been included in the discussion. I agree that not excluding preeclamptic mothers can contribute to increased ratio of SGA infants (depending on maternal ratio, of course). Simple, minor changes will enhance readability further.

Line 79: "of Caucasian" can be changed to "Caucasian" or "of Caucasian origin".

Line 109: "and" can be omitted between length, head circumference.

Line 112: "," can be added after total hospital days (in days).

7. PLOS authors have the option to publish the peer review history of their article (what does this mean? ). If published, this will include your full peer review and any attached files.

**Do you want your identity to be public for this peer review?** For information about this choice, including consent withdrawal, please see our Privacy Policy .

Reviewer #1: No

Reviewer #2: No

---

## [Author Response · Author response to Decision Letter 2]

2 Apr 2025

Reviewer #2: The authors have taken good efforts to address most of the issues. Reasoning behind results have been included in the discussion. I agree that not excluding preeclamptic mothers can contribute to increased ratio of SGA infants (depending on maternal ratio, of course). Simple, minor changes will enhance readability further.

Line 79: "of Caucasian" can be changed to "Caucasian" or "of Caucasian origin".

This sentence has been modified in accordance with your suggestion.

“…whose parents were Caucasian and born in Spain”

Line 109: "and" can be omitted between length, head circumference.

We have corrected this sentence based on the feedback provided.

“The following variables were analyzed: gestational age, birth weight, length, head circumference, Apgar score, bronchopulmonary dysplasia (BPD)…”

Line 112: "," can be added after total hospital days (in days).

This line has been revised according to your comment.

“…sepsis, total NICU stay (in days), days of mechanical ventilation, total hospital stay (in days), and mortality.”

---

## [Editor Report · Decision Letter 3]

9 Apr 2025

Classification of very low birth weight infants as small for gestational age: International vs. national standards.

PONE-D-24-41520R3

Dear Dr. Marta Tejón Fernández,

We’re pleased to inform you that your manuscript has been judged scientifically suitable for publication and will be formally accepted for publication once it meets all outstanding technical requirements.

Kind regards,

Tamara Sljivancanin Jakovljevic

Academic Editor

PLOS ONE

---

## [Editor Report · Acceptance letter]

PONE-D-24-41520R3

PLOS ONE

Dear Dr. Tejón-Fernández,

I'm pleased to inform you that your manuscript has been deemed suitable for publication in PLOS ONE. Congratulations! Your manuscript is now being handed over to our production team.

Kind regards,

on behalf of

Dr. Tamara Sljivancanin Jakovljevic

Academic Editor

PLOS ONE